# Mixed and Multi-Methods Protocol to Evaluate Implementation Processes and Early Effects of the Pradhan Mantri Jan Arogya Yojana Scheme in Seven Indian States

**DOI:** 10.3390/ijerph17217812

**Published:** 2020-10-26

**Authors:** Manuela De Allegri, Swati Srivastava, Christoph Strupat, Stephan Brenner, Divya Parmar, Diletta Parisi, Caitlin Walsh, Sahil Mahajan, Rupak Neogi, Susanne Ziegler, Sharmishtha Basu, Nishant Jain

**Affiliations:** 1Heidelberg Institute of Global Health, Medical Faculty and University Hospital, Heidelberg University, 69120 Heidelberg, Germany; swati.srivastava@uni-heidelberg.de (S.S.); Stephan.brenner@uni-heidelberg.de (S.B.); Diletta.parisi@uni-heidelberg.de (D.P.); cmwalsh17@gmail.com (C.W.); 2German Development Institute/Deutsches Institut für Entwicklungspolitik (DIE), 53113 Bonn, Germany; Christoph.Strupat@die-gdi.de; 3Centre for Global Health and Health Partnerships, School of Population Health and Environmental Sciences, King’s College London, London SE5 9RJ, UK; divya.parmar@kcl.ac.uk; 4IQVIA Consulting and Information Services India, New Delhi 110001, India; smahajan.spj@gmail.com; 5Nielsen India Private Limited, Gurugram 122002, India; rupak.neogi@nielsen.com; 6Indo-German Social Security Programme (IGSSP), Deutsche Gesellschaft für Internationale Zusammenarbeit (GIZ) GmbH, New Delhi 110029, India; susanne.ziegler@giz.de (S.Z.); sharmishtha.basu@giz.de (S.B.); nishant.jain@giz.de (N.J.)

**Keywords:** health insurance, health financing, India, impact assessment, supply-side, demand-side, process evaluation

## Abstract

In September 2018, India launched Pradhan Mantri Jan Arogya Yojana (PM-JAY), a nationally implemented government-funded health insurance scheme to improve access to quality inpatient care, increase financial protection, and reduce unmet need for the most vulnerable population groups. This protocol describes the methodology adopted to evaluate implementation processes and early effects of PM-JAY in seven Indian states. The study adopts a mixed and multi-methods concurrent triangulation design including three components: 1. demand-side household study, including a structured survey and qualitative elements, to quantify and understand PM-JAY reach and its effect on insurance awareness, health service utilization, and financial protection; 2. supply-side hospital-based survey encompassing both quantitative and qualitative elements to assess the effect of PM-JAY on quality of service delivery and to explore healthcare providers’ experiences with scheme implementation; and 3. process documentation to examine implementation processes in selected states transitioning from either no or prior health insurance to PM-JAY. Descriptive statistics and quasi-experimental methods will be used to analyze quantitative data, while thematic analysis will be used to analyze qualitative data. The study design presented represents the first effort to jointly evaluate implementation processes and early effects of the largest government-funded health insurance scheme ever launched in India.

## 1. Introduction

In line with global efforts aimed at leaving no one behind while advancing universal health coverage (UHC), recent health sector reforms in India have strived to expand social health protection for the most vulnerable segments of society while simultaneously strengthening delivery of quality healthcare services. This strategy has largely been pursued through the implementation of government-funded health insurance (GFHI) schemes implemented at either federal or state levels [1], aimed at improving access to quality health care while offering financial protection to end users. The expansion of these schemes has led to a steady rise in population insurance coverage, from approximately 5% of households in 2005 to 29% in 2015 [2].

Evidence on the effect of these Indian GFHI schemes on health service use, financial protection, quality of service provision, and health outcomes presents a diverse view. A recent systematic review on the impact of Indian GFHI schemes found positive effects on health service use but mixed effects on out-of-pocket expenditures (OOPEs) and financial risk protection [3]. Only one study demonstrated better health outcomes in enrolled versus non-enrolled households for covered conditions [4]. The few studies on gender effects of Indian GFHI schemes report the persistence of lower access, enrolment, and use among women than men [5,6,7,8,9]. There is little evidence on the effect of in-built quality control mechanisms in improving the quality of service provision [10,11,12]. Likewise, little is known on scheme implementation processes and their role in shaping impacts [13,14,15,16]. Even fewer studies explore factors relating to scheme ideation, design, and conceptualization [17]. A review of numerous impact studies on one specific Indian scheme, Rashtriya Swasthya Bima Yojana (RSBY), highlights methodological weaknesses in existing studies and calls for more rigorous evaluations [18]. This paucity of evidence is surprising considering that the scope of GFHI implementation in the Indian context can offer numerous opportunities for learning at the global level. 

Our study is situated against this background and aimed to generate scientific evidence on both implementation processes and early effects of the largest GFHI scheme in India, the Pradhan Mantri Jan Arogya Yojana (PM-JAY). The ambition of our study, which combines process and impact evaluation elements to look at the first 18 months of operation, was to generate evidence to guide future implementation of the scheme while simultaneously drawing lessons to inform social health protection developments in other settings. 

Hereafter, we first describe the scheme and its complexities and then present details of our study design, including both the conceptual model and the methodology guiding our evaluation work.

## 2. Materials and Methods 

### 2.1. Pradhan Mantri Jan Arogya Yojana

In September 2018, the Government of India (GoI) launched PM-JAY [19]. The GoI expects PM-JAY to consolidate gains made by previous and ongoing GFHI schemes, primarily RSBY, by removing formal enrolment procedures and expanding coverage to approximately 500 million people listed as most vulnerable in the Socio-Economic Caste Census (SECC) of 2011 [20]. Insurance coverage under this new scheme is no longer subject to a fixed household size but instead covers year INR 500,000 (EUR 6150 in May 2020) per household with no member ceiling; the benefit package includes 1350 inpatient procedures provided at secondary and tertiary care levels. Beneficiaries can avail these services at any public and empaneled private hospital across all Indian states implementing PM-JAY. The scheme is governed by the National Health Authority (NHA) at the federal level and implemented by the State Health Agency (SHA) in each state, either through existing insurance companies, or publicly run autonomous trusts/nodal agencies, or a combination of these two (referred to as mixed models), with a scope for state-specific adjustments. Currently, 32 out of 36 states and union territories in India are implementing the scheme [21]. PM-JAY seeks both to expand health care access and financial protection, and to promote efficiency, transparency, and accountability by adopting output-based financing measures. 

PM-JAY implementers ultimately expect the scheme to increase affordable access to quality healthcare services for the most vulnerable segments of society. They anticipate the scheme to produce significant changes on all three UHC dimensions: population coverage (through comprehensive eligibility and removal of any formal enrolment procedure for the most vulnerable population segments); service coverage (through a generous benefit package covering high-cost services offered through a wide network of PM-JAY-accredited and empaneled providers); and financial protection (through cashless provision and high insurance ceiling, reducing out-of-pocket payments). 

### 2.2. Conceptual Framework and Study Objectives

In order to assess the effect of PM-JAY on outcomes related to health and healthcare, we rely on a conceptual framework (Figure 1), which explicitly acknowledges access to healthcare services as the product of the interaction between supply- and demand-side factors [22]. 

Our study design aligns with this conceptual postulation and explores both demand- and supply-side elements shaping population coverage first, and access to care and financial protection afterwards. In addition, our design recognizes the socio-political dimension of policy implementation as an element that affects intervention outcomes by including a policy-level process documentation aimed at disentangling how the overall policy context and implementation realities ultimately shape behavior, both on the demand and on the supply sides. 

The study objectives were:To assess and understand the effects of PM-JAY on the target population’s awareness of PM-JAY features, coverage, health service use, and financial protection, including unraveling effect heterogeneity by population sub-groups (e.g., along socio-economic and gender dimensions);To assess the effects of PM-JAY on quality of service delivery and to explore healthcare providers’ experiences with the implementation of PM-JAY; andTo document implementation processes at the national level, as well as in selected states transitioning from either no or a prior GFHI to PM-JAY.

Each study objective corresponds to a component within the study design (Figure 2). Study components 1 and 2 look respectively at the effects of the scheme on the demand and on the supply of secondary and tertiary care services. Study component 3 documents the process of designing and implementing PM-JAY.

### 2.3. Study Design

This study adopts a concurrent triangulation mixed methods design [23] (Figure 3), relying on a multi-method approach [24]. The term ‘mixed methods’ is used to refer to the inclusion of both quantitative and qualitative methods of data collection and analysis. The term ‘multi-method’ refers to the fact that our design incorporates diverse methods of data collection and analysis within each strain of research, i.e., qualitative and quantitative [24]. Specifically, in a triangulation design, quantitative and qualitative methods hold equal weight and are used to explore different facets of the same phenomenon [23]. In our specific case, we selected triangulation above any other mixed-methods design because we focus on different facets of a single phenomenon, the implementation and effects of the PM-JAY. We used quantitative methods in a quasi-experiment to quantify scheme awareness and population coverage among the target community and to assess the scheme effect on outcomes of interest (health service use and financial protection); we used qualitative methods to explore causal mechanisms of change through an analysis of implementation processes. The term ‘concurrent’ refers to the fact that quantitative and qualitative data are collected in parallel, without waiting for the findings from one strain of research to be available to inform data collection for the other strain [23]. The design was set exclusively by the authors. 

### 2.4. Study Setting and State and District Selection

The study took place in seven Indian states, namely Bihar, Chhattisgarh, Gujarat, Karnataka, Meghalaya, Tamil Nadu, and Uttar Pradesh. Both states and districts within states were purposively selected by policy stakeholders in consultation with study funders (the Deutsche Gesellschaft für Internationale Zusammenarbeit (GIZ) GmbH) to reflect the geographical diversity and landscape of the earlier RSBY as well as the diversity of implementation models within PM-JAY itself: one state (Meghalaya) is implementing the scheme through insurance companies, three states (Bihar, Karnataka, and Uttar Pradesh) are implementing the scheme though state agencies, and the remaining three states (Chhattisgarh, Gujarat, and Tamil Nadu) are implementing the scheme through mixed models involving both state agencies and insurance companies. While prior to PM-JAY, Chhattisgarh and Meghalaya states had successfully used the RSBY platform to extend coverage to nearly their entire population via additional state funding, the states of Gujarat, Karnataka, and Tamil Nadu had successfully implemented their own state-funded tertiary care schemes. Bihar and Uttar Pradesh are two states in which RSBY implementation had already been discontinued for several years. Similarly, two districts were selected in each state (three districts in Bihar and Uttar Pradesh) to account for diverse performance levels within states (Table 1 and Figure 4). 

Hereafter, we describe each of the study components in detail.

### 2.5. Study Component 1: Demand-Side Household and Individual Study

This component integrates quantitative and qualitative methods to address the first study objective. A single data collection round was planned for the period November 2019 to May 2020. Data collection was suspended starting in late March 2020 due to the SARS-CoV-2 epidemic.

#### 2.5.1. Quantitative Component

##### Sampling 

Our quantitative sampling strategy aims to include both a representative sample of PM-JAY eligible households at the district level and a sufficiently representative sample of those eligible households having advanced a claim under PM-JAY by the time of data collection, approximately 14 to 16 months after the scheme launch. Within all the above-mentioned districts, our primary sampling unit (PSU) included villages in rural areas and medium/small urban areas with less than 7500 inhabitants (based on the observations of our pilot surveys, we set the cutoff at 7500 inhabitants to exclude larger urban areas in which administering the survey was not operationally feasible). Sampling adopted a three-step approach. First, information from the 2011 SECC was used to randomly select 50 PSUs that were representative of the entire district with regards to the following district characteristics: number of PM-JAY eligible households, number of PM-JAY eligible individuals, number of PM-JAY claims, share of urban households, share of women, share of children, share of adults, share of elderly, and share of married people. Second, using the selected PSUs and SECC information, 15 households per PSU were randomly selected to reach a total representative sample of 750 eligible households per district. Third, 20 additional PSUs were identified among those with the highest number of claims across all remaining PSUs in a district. In order to ensure a sufficient sample of households with a filed claim under PM-JAY, 20 claim households were selected in each of these 20 PSUs, to reach a total sample of 400 claim households per district. This strategy aimed to reach 1150 households per district, for a total of 18,400 households across the seven states. 

##### Data Collection Tools and Strategies

Data were collected via a household survey containing questions on household socio-demographics, household loans and consumption expenditure, illness reporting (for acute and chronic illnesses and related inpatient hospitalizations), health service use, out-of-pocket spending on health, and PM-JAY awareness and experiences. Given the scheme’s focus on secondary and tertiary care, the survey included questions on inpatient hospitalizations (both surgical and non-surgical) over the prior 12 months. A specific module was built within the survey to assess women’s experiences with the scheme and its role in mediating household decision-making and enhancing women’s empowerment. Data were to be collected by trained interviewers relying on the computer-assisted personal interviewing (CAPI) on digital devices. The survey was translated in all concerned local languages and administered by local interviewer teams perfectly fluent in these languages and recruited directly within the study districts. 

##### Analytical Approach

Descriptive statistics will be used to describe awareness of the scheme and its operational modalities. The scheme effects will be assessed by establishing changes on reported hospitalization rates and OOPE (defined as primary outcomes in line with the scheme focus) by applying a difference-in-difference framework (DID). Using data on the exact empanelment dates of all hospitals in the survey districts and by using the detailed information on hospitalization cases (including date and place of hospitalization) from the household survey, we will be able to compare hospitalization and OOPE of individuals at different points in time (from the date of hospitalization) that are living in districts where many hospitals were empaneled immediately after the start of PM-JAY (treatment group) with districts where the rollout of PM-JAY and the empanelment of hospitals took more time (control group). In order to control for time-invariant state characteristics, such as health infrastructure, that are likely to be correlated with both the hospital empanelment dates in the districts and our primary outcome variables, we will include state dummies into all our specifications. Furthermore, we will include individual and household-specific variables in our specifications.

#### 2.5.2. Qualitative Component

##### Sampling

Qualitative data collection also took place across the abovementioned districts, specifically in a sub-set of communities targeted by the quantitative survey, and relied on a combination of focus group discussions (FGDs) among PM-JAY eligible community members, individual in-depth interviews (IDIs) with individuals who had used services under PM-JAY, and key informant interviews (KIIs) among community health workers (CHWs), as those in charge of PM-JAY community dissemination and support activities. We combined FGD and IDI because while the former are useful to elicit knowledge on overall community attitudes, beliefs, and social constructions, the latter are useful to gain in-depth insight into individual views and experiences of PM-JAY [25].

Although settling on a specific sample size is beyond the scope of qualitative research [25], we expected to conduct a minimum of 2 FGD in each district, 1 in a rural and 1 in an urban setting (*n* = 2 × 16 districts = 32 FGD), with 8–10 participants in each FGD, specifically selecting PM-JAY eligible individuals with or without PM-JAY service use experience. Given the specific cultural context within which our study takes place as well as our wish to maintain a specific focus on gender dynamics throughout the study, FGDs will either be all female or all male. From each FGD, we will select 1–2 information-rich cases, i.e., individuals whose experience with PM-JAY service use appears to be particularly poignant in relation to the underlying research questions (for example those who have experienced high-cost treatments or have lodged grievances under PM-JAY) and follow them up with a request to engage in an additional IDI. We expect to conduct a total of 60–80 IDIs. In addition, we plan to interview at least two CHWs per district as key informants, providing information on the overall scheme rollout and helping us to identify respondents for the FGD and the IDIs.

##### Data Collection Tools and Strategies

FGDs, IDIs, and KIIs are to be conducted in the local state-specific language, recorded after obtaining written informed consent, transcribed verbatim, and translated into English for analysis. Data collection is to be carried out by trained qualitative research assistants, using the guides briefly described hereafter. 

FGD, IDI, and KII discussion guides include themes on health conditions requiring treatment, provider availability in the community including perceptions around their quality, arranging finances for healthcare, household decision-making processes for healthcare and changes therein, awareness of PM-JAY, awareness about eligibility, knowledge about PM-JAY, experiences with utilizing services under PM-JAY (including grievances), and review/assessment of PM-JAY. The IDI guide will also explicitly focus on the illness and service utilization experience and gender-related aspects of the main themes, while the KII guide will also explore PM-JAY-related roles and responsibilities of CHWs.

##### Analytical Approach

Transcripts from FGD, IDI, and KII will be analyzed using the framework approach [26,27], relying on a mixed deductive and inductive approach [25]. We will develop an initial set of coding categories based on the themes of the interview guides but will allow additional codes and themes to emerge iteratively. We will apply source and analyst triangulation [25]. Data analysis will be conducted using Nvivo Pro 12 QSR International Software (QSR International, Doncaster, Australia) [28].

### 2.6. Study Component 2: Supply-Side Hospital Evaluation

This component integrates quantitative and qualitative methods to addresses the second study objective. Data collection was carried out during two repeated rounds (the first from February to April 2019 and the second from November 2019 to January 2020).

#### 2.6.1. Quantitative Component

##### Sampling

This study component included two different samples, a facility sample and a patient sample. For the facility sample, we identified a non-random sample of 6 empaneled (either in PM-JAY scheme or in a state GFHI) and 6 non-empaneled hospitals in each of 14 of the abovementioned 16 districts (the supply-side hospital evaluation was not conducted in two districts: Gaya (Bihar) and Rampur (Uttar Pradesh). In each district, empaneled and non-empaneled hospitals were balanced (but not matched) on the following characteristics: rural vs. urban setting, hospital size, public vs. private ownership, and medical specialties offered. Information on hospital characteristics were obtained from PM-JAY district nodal officers. The resulting total sample size included 168 hospitals across 14 districts and 7 states. For the patient sample, 6–8 discharged patients previously hospitalized either under PM-JAY or another state-specific insurance scheme were randomly selected from all discharged patients at each sampled hospital at the day of data collection.

##### Data Collection Tools and Strategies

This study component included three quantitative surveys applied across sampled hospitals: (1). a checklist on hospital infrastructure (i.e., information on facility accessibility, availability of service units and basic equipment, health management and referral structures, safety and monitoring measures, transportation means); (2). a checklist on health service and administrative processes (i.e., information on aspects related to empanelment and accreditation, extent to which empanelment criteria are met, extent to which scheme-specific processes (e.g., trainings, claims settlement, monitoring, reporting, audits, etc.) are met); and (3). exit interviews with previously hospitalized patients (i.e., information on socio-demographic aspects, health insurance coverage, hospitalization and pre-hospitalization aspects (e.g., referral, OOPE, services received, inpatient accommodation, and amenities), personal experience, and perception of hospital stay). 

Trained research assistants using CAPI digitalized solutions will carry out data collection using the three tools. The two checklists on infrastructure and service/administrative processes are to be completed by the hospital director or manager with research assistants directly observing selected items and/or processes.

##### Analytical Approach 

Variables obtained by these three quantitative surveys will be examined and combined into composite indicators measuring the following key areas: general availability and readiness of hospital, specific availability and readiness of empaneled secondary level services (i.e., medical, surgical, obstetric, emergency), implementation of scheme-relevant processes, and patient satisfaction with received services. Composite indices will be developed based on a conceptional framework matrix reflecting both the World Health Organization’s Service Availability and Readiness Assessment criteria [29] as well as PM-JAY hospital implementation guidelines [30]. The resulting composite indicators will serve as primary outcome measures in measuring both scheme effects and implementation progress.

Since our sample involves both PM-JAY empaneled and non-empaneled (i.e., control) hospitals within the same study districts, we propose a quasi-experimental approach to impact estimation based on a DID evaluation of hospitals across the two data collection rounds: empaneled hospitals are considered as “treatment” and the remaining as “comparison”. By comparing pre- and post-PM-JAY quality of care outcomes in treatment and comparison hospitals, we can capture the causal impacts of PM-JAY. We may further ensure comparability of treatment (empaneled) and comparison (not-empaneled) hospitals by employing coarsened exact matching [31] methods to control for hospital observable covariates, such as bed strength, human resources, infrastructure, supplies, and others.

#### 2.6.2. Qualitative Component

##### Sampling

Qualitative data sources considered for this study component included at least one individual service provider from each empaneled and non-empaneled hospital. These individuals include the main hospital administrator/manager (i.e., the hospital director in most instances) and one key clinical provider (i.e., ideally the provider in charge of general admission of PM-JAY patients, usually a senior general physician). In non-empaneled hospitals, only the main hospital administrator/manager will be interviewed, as their clinical providers are not expected to deal with PM-JAY patients. The resulting qualitative sample will therefore include approximately 250 individuals.

##### Data Collection Tools and Strategies

Qualitative information is collected through IDI. We will use three different interview guides depending on hospital empanelment status and respondent background. For IDI with administrators/managers at empaneled hospitals, interviews focus on experiences and perceptions of the PM-JAY implementation process in relation to managerial/administrative aspects of providing quality services and resulting suggestions for further improvements in PM-JAY implementation. For IDI with key clinical providers at empaneled hospitals, interviews focus on experiences and perceptions of PM-JAY and other GFHI implementation processes in relation to clinical aspects of providing quality services and resulting suggestions for further improvements in PM-JAY implementation. For IDI with administrators/managers at non-empaneled hospitals, interviews focus on experiences and perceptions of the implementation process in relation to managerial/administrative aspects of providing health care services, challenges or experiences with past empanelment in PM-JAY or other GFHI, and suggestions for these schemes, with an additional focus on why these hospitals did not decide to empanel under PM-JAY.

Data collection will be carried out during two repeated rounds, at the same time as the hospital surveys by trained research assistants. 

##### Analytical Approach 

The analytical approach will be the same as for the qualitative component in study component 1.

### 2.7. Study Component 3: Process Documentation

This study component adopts a qualitative approach to address the third research question by reconstructing and documenting processes related to the ideation, development, and implementation of PM-JAY at the national and state levels. Rooted in a constructivist perspective approach [32], we explored PM-JAY early implementation through the eyes of the people directly involved in the scheme design and rollout.

#### 2.7.1. Sampling 

Data for this component was collected in Chhattisgarh, Tamil Nadu, and Uttar Pradesh. These states were purposely selected by scheme implementers to reflect the diversity of PM-JAY implementation modes. In addition, we collected data in the capital city, Delhi, to capture the views of national-level policy makers and implementers. At the state level, we included individuals from the apex scheme implementation body (the State Health Agency or analogous entity within the state); technical support units within the SHA, such as consultant organizations or information technology teams; implementation support agencies, such as insurance companies or third-party administrators; and any external organizations providing independent technical or managerial support in scheme implementation. At the district level, we included district functionaries directly involved in the implementation of PM-JAY. In addition, at the national level, we used the snowball sampling approach to identify respondents from all key stakeholder groups involved in PM-JAY implementation and development, i.e., both closely involved actors (such as government officials and policy makers, international, bi- and multi-lateral agencies, and PM-JAY implementers) and more distantly involved groups (such as academia and civil society).

#### 2.7.2. Data Collection Tools and Strategies

Data collection relies on KIIs conducted with one of three interview guides developed by the study team, with one guide focusing on policy makers, one on state-level functionaries, and one on district implementers. In each of the three guides, we explore the fidelity of implementation to the original design and related adaptations; changes to administrative structures and authority within governing institutions; changes in implementation processes; how stakeholders responded to implementation; and any unintended consequences arising from implementation processes. Data collection in Delhi aimed at capturing national dynamics related to ideation, planning, and implementation; data collection at state and district levels aimed at capturing state-specific implementation realities.

Policy-level interviews were conducted either by the principal investigator and/or by two additional senior researchers (all authors). Interviews at the district level were conducted by a small team of senior qualitative researchers. 

#### 2.7.3. Analytical Approach

Analysis of early implementation developments is considered an ongoing process that requires integration of the implementation science perspective [33] into the political economy context [34]. Thus, using thematic analysis, we will describe PM-JAY implementation processes, including achievements and challenges, in relation to their wider socio-political context within which the scheme implementation takes place. This approach entails that at different analytical levels, we aim at understanding how the new scheme is shaped by existing authority structures but at the same time also challenged by these authority structures. 

### 2.8. Ethical Considerations

Ethical approval for the study was obtained from the Ethikkommission of the Medizinische Fakultät Heidelberg, Germany, and from Sigma Institutional Review Board, New Delhi, India via IRB numbers 10057/IRB/19-20 (study component 1) and 1011/IRB/D/18-19 (study components 2 and 3). Written informed consent was sought from all study participants.

### 2.9. Patient and Public Involvement

The public was not involved in study design or data collection, and will not be involved in analysis. 

### 2.10. Data Sharing

No additional data are available as this is a study protocol and no data are being used. The data produced from this study will not be available for open access, but the corresponding author can be contacted in case of any queries.

## 3. Discussion

The study design presented in this article represents the first effort to jointly evaluate the implementation processes and early effects of the largest GHFI scheme ever launched in India, PM-JAY. Describing our study protocol serves the purpose of highlighting the efforts channeled towards generating sound scientific evidence in light of real-life evaluation needs, determined by policy considerations and implementation realities. Given PM-JAY’s national reach, it was impossible for our team to control elements related to treatment assignment and rely on a standard experimental design for evaluation. Similarly, decisions on state and district sampling were made a priori, in light of specific policy needs, constraining the team’s methodological choices. Notwithstanding such challenges, we do trust having been able to produce a sound study design, which draws its strengths primarily from the mixed- and multi-method nature of its data collection and analysis effort and from the ability to identify and integrate quasi-experimental design elements in the quantitative analytical strain. We started with the scheme implementation reality and co-created a study design that could accommodate it while at the same time adhering to best research practices.

A further strength of our study design rests in the fact that elements related to process and impact evaluation are embedded within a single study protocol, addressing both demand- and supply-side implementation realities and early effects. This integration of multiple research strains (both from a content and from a methodological perspective) within a single study protocol favors triangulation since it assigns responsibility for a very diverse set of analytical components to a single study team. While data collection activities are fixed, and further constrained by the challenges of the SARS-CoV-2 epidemic at the time of writing this article, reliance on a single team allows analysis to proceed in an emergent manner [35], with process elements being used to elucidate effect results and effect results being used to inspire further investigation into process elements.

Notwithstanding the potential to yield informative results through a very enriching analytical process, we must acknowledge the complexity of our study design and the challenges inherent in leading our combined process and impact evaluation towards a successful end. First, implementing our study design requires reliance on a very diverse team of researchers, with very different content and methodological expertise. It follows that while ultimately enriching, the process of integrating findings across analytical strains will inevitably be difficult and possibly lead to occasional conflict. Second, the breadth of our design, and of the different elements it encompasses, will inevitably add depth to our work but also complexity to our dissemination efforts. The team will have to find ways to present results in a clear and effective manner, drawing from multiple analytical strains to ensure that the broader policy and academic community embrace the complexity of scheme implementation and effects. Given the attention that efforts to expand social health protection worldwide are receiving, our team will need to have the capacity to ensure adequate dissemination well beyond Indian boundaries. 

In addition, we must acknowledge the limited external validity of our study design [36]. Not only were states and districts within states purposely selected ex-ante, but India is a vast country characterized by the co-existence of very different realities. Hence, while we trust that our overall approach to sampling and data collection yields a high internal validity [36], we can by no means assert that our findings will provide an accurate picture of PM-JAY implementation processes and early effects for the entire country. In addition, we must acknowledge the inherent bias that arises from the sampling strategies adopted both to identify households for the demand-side component and hospitals for the supply-side component. Prior to the beginning of our study, we pondered multiple alternatives and convened that in the absence of a more recent complete population listing and in the absence of a complete health facility listing (or of resources to undergo one on our own), we had no alternative but to adopt the strategies described earlier in this manuscript. With specific reference to the demand-side component, we must acknowledge that our findings will not be illustrative of urban realities. Further research, relying on different primary sampling frames, is needed to explore implementation realities and early effects in urban settings. 

## 4. Conclusions

The study design presented represents the first effort to jointly evaluate implementation processes and early effects of the largest government-funded health insurance scheme ever launched in India. We hope that having described our study design in detail will provide inspiration for other researchers engaging in analysis of emerging social health protection schemes in India and in the broader region.

## Figures and Tables

**Figure 1 ijerph-17-07812-f001:**
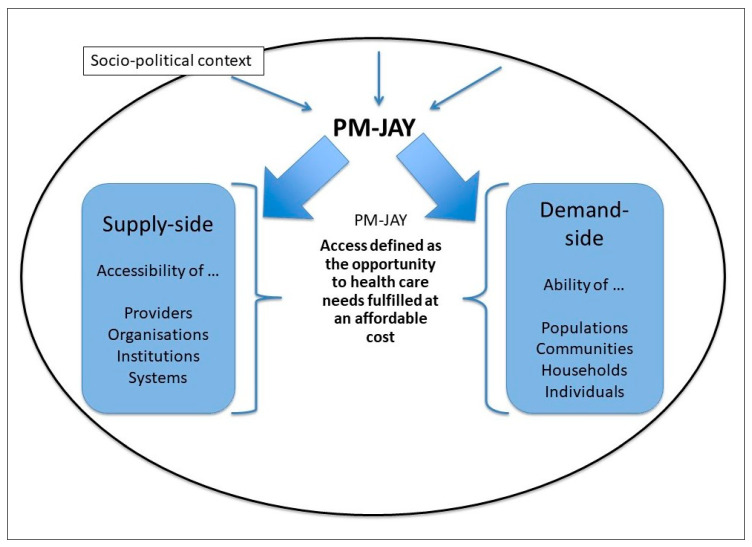
Conceptual framework, adapted from Levesque et al., 2013 [22].

**Figure 2 ijerph-17-07812-f002:**
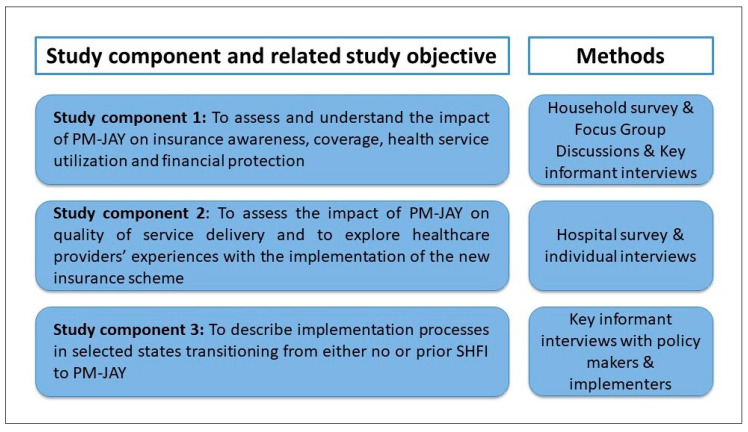
Study components, study objectives, and methods.

**Figure 3 ijerph-17-07812-f003:**
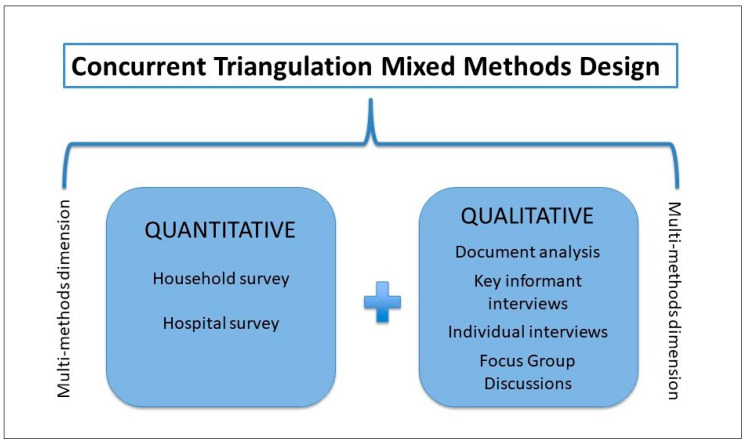
Study design.

**Figure 4 ijerph-17-07812-f004:**
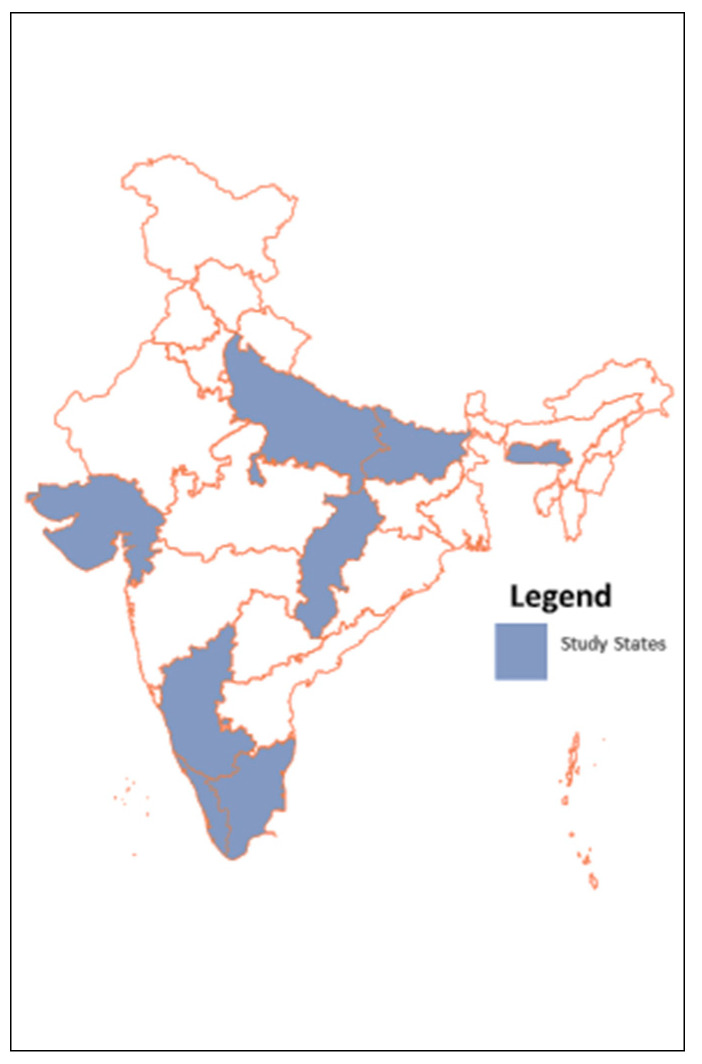
Study states.

**Table 1 ijerph-17-07812-t001:** Study states, implementation modes, and districts.

State	Implementation Model	Districts	Previously Implemented GFHI in 2018
Bihar	Trust mode	Gaya, Muzaffarpur and Patna	None
Chhattisgarh	Mixed mode *	Raigarh and Bilaspur	Mukhyamantri Swasthya Bima Yojana
Gujarat	Mixed mode	Ahmedabad and Surat	Mukhyamantri Amrutam
Karnataka	Trust mode	Raichur and Tumkur	Vajpayee Arograsri Scheme
Meghalaya	Insurance mode	South West Garo Hills and East Khasi Hills	Megha Health Insurance Scheme
Tamil Nadu	Mixed mode	Coimbatore and Sivagangai	Chief Minister’s Comprehensive Health Insurance Scheme
Uttar Pradesh	Trust mode	Allahabad, Ghazipur and Rampur	None

* At the time of study conceptualization, Chhattisgarh operated as a mixed mode; as of October 2019, it transitioned to trust mode.

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
