# Peer review of "Mixed and Multi-Methods Protocol to Evaluate Implementation Processes and Early Effects of the Pradhan Mantri Jan Arogya Yojana Scheme in Seven Indian States"

_ijerph, 2020, doi:10.3390/ijerph17217812_

Round 1

Reviewer 1 Report

Thank you for letting me review the paper “Mixed and multi-methods protocol to evaluate implementation processes and early effects of the Pradhan Mantri Jan Arogya Yojana scheme in seven Indian States”. The paper describes the methodology adopted to evaluate the PM-JAY.

The paper is an impressive description of an important aspect of modern-day life; health care and health insurance for all. Overall, I think the methodology described is massively impressive and the evaluation of the plan is not only thorough but also very intricate with many different aspects covered. Since the paper is a description of strategies and not a fully implemented model one must remember that review comments are meant as considerations for the full implementation.

I regard to the quantitative components sampling strategy, the overall idea and methodology is sound. I find the 10% sampling strategy in the first component ambitious and but I might be missing something important; if 50 PSU’s are selected that are representative and 15 are randomly selected I reach 75 households and not 750 as described. I think a 1% sample might be too small, but I understand that the correct numbers might be 500 and 150.

I find no information in regard to non-reply in the survey of both components. If some of the rural areas are deprived, I suspect that a substantial number of surveys might end up not being replied. I think an important addition to the sampling strategy is a strategy to handle missing data – will the authors add respondents if some fail to answer? Will they accept smaller sample sizes in some areas and what are the criteria?

DID-methods are valuable in some sense, but I think the authors should consider models that can better encompass issues of selection into different areas. This is more of a suggestion than something I think MUST be added to the paper; the authors could consider applying treatment effects methods like probability weighted regression adjustments to better account for the selection bias in both usage/treatment but also in place of living.

Regarding the qualitative sampling, I think this seems less ambitious but perhaps adequate. When reading the very ambitious quantitative sampling strategy, a total of 32 interviews in the first component seems on the low side and the authors could consider perhaps a second wave after the initial results from the quantitative analysis to better encompass interesting findings – in that way, the quantitative analysis works as a guide for the qualitative interviews and heightens to coherence between the studies.

Overall I find the paper very interesting and I think not only the PM-JAY but also the future results from this evaluation plan to be important and interesting.

Author Response

Response to Reviewer 1 Comments

Thank you for letting me review the paper “Mixed and multi-methods protocol to evaluate implementation processes and early effects of the Pradhan Mantri Jan Arogya Yojana scheme in seven Indian States”. The paper describes the methodology adopted to evaluate the PM-JAY.

The paper is an impressive description of an important aspect of modern-day life; health care and health insurance for all. Overall, I think the methodology described is massively impressive and the evaluation of the plan is not only thorough but also very intricate with many different aspects covered. Since the paper is a description of strategies and not a fully implemented model one must remember that review comments are meant as considerations for the full implementation.

I regard to the quantitative components sampling strategy, the overall idea and methodology is sound. I find the 10% sampling strategy in the first component ambitious and but I might be missing something important; if 50 PSU’s are selected that are representative and 15 are randomly selected I reach 75 households and not 750 as described. I think a 1% sample might be too small, but I understand that the correct numbers might be 500 and 150.

Response 1: We thank the reviewer for their comments. In each district, there are a total of 50 PSUs randomly selected from the SECC eligibility list, with approximately 15 households per PSU (totaling 50 X 15 = 750 households). IN each district, we also sample 20 additional PSUs from claim-advancing households, with approximately 20 households in each of these PSUs (totaling 20 X 20 = 400 households). The total number of households for each district is then 750 + 400 = 1,150.

I find no information in regard to non-reply in the survey of both components. If some of the rural areas are deprived, I suspect that a substantial number of surveys might end up not being replied. I think an important addition to the sampling strategy is a strategy to handle missing data – will the authors add respondents if some fail to answer? Will they accept smaller sample sizes in some areas and what are the criteria?

Response 2: For all districts, we had a sufficient list of replacement households for households from the SECC eligibility lists. For claims households we sampled nearly all claim households within the PSU. In PSUs with more than 20 claims, we used the remaining households after the sampling as replacement households in case the sampled households were not traceable.

DID-methods are valuable in some sense, but I think the authors should consider models that can better encompass issues of selection into different areas. This is more of a suggestion than something I think MUST be added to the paper; the authors could consider applying treatment effects methods like probability weighted regression adjustments to better account for the selection bias in both usage/treatment but also in place of living.

Response 3: We are thankful for this methodological suggestion and will consider it at time of analysis.

Regarding the qualitative sampling, I think this seems less ambitious but perhaps adequate. When reading the very ambitious quantitative sampling strategy, a total of 32 interviews in the first component seems on the low side and the authors could consider perhaps a second wave after the initial results from the quantitative analysis to better encompass interesting findings – in that way, the quantitative analysis works as a guide for the qualitative interviews and heightens to coherence between the studies.

Response 4: We are aware that the qualitative sample within each district is relatively small, but we are constrained by resources related to the need to conduct work across seven States. Therefore, we have to keep the sample on the lower side in each district. We also note that 32 refers only to the basic number of expected FGD with community members. In addition, in each district, we have individual interviews with beneficiaries and key informant interviews with stakeholders.

Overall I find the paper very interesting and I think not only the PM-JAY but also the future results from this evaluation plan to be important and interesting.

Reviewer 2 Report

The work seems to me overall interesting and kind of hot topic in present times, but there is a clue I actually do not understand. In the conclusions you say that the fragmentation and huge differences among provinces (or whatever you want to call the local entities) do not guarantee the application of the model in the whole country. This let me rise a question about the criteirion (or criteria) you picked in order to select the provinces object of your study. Would you please clarify it? If not mentioned, I guess it could be a crucial flaw in your study, and I am not here to reject your work, so please provide a solid explanation to this issue.  

Author Response

Response to Reviewer 2 Comments

The work seems to me overall interesting and kind of hot topic in present times, but there is a clue I actually do not understand. In the conclusions you say that the fragmentation and huge differences among provinces (or whatever you want to call the local entities) do not guarantee the application of the model in the whole country. This let me rise a question about the criteirion (or criteria) you picked in order to select the provinces object of your study. Would you please clarify it? If not mentioned, I guess it could be a crucial flaw in your study, and I am not here to reject your work, so please provide a solid explanation to this issue.  

Response 1: We thank the reviewer for their comment. The states and districts selected for the study were selected by policy stakeholders in consultation with the study funders and described in lines 154-174, under the section “Study setting and State and district selection”. States were selected to reflect the geographical diversity and landscape of the earlier RSBY as well as the diversity of implementation models within PM-JAY itself: one state (Meghalaya) was implementing the scheme through insurance companies, three states (Bihar, Karnataka and Uttar Pradesh) are implementing the scheme though state agencies, and remaining three states (Chhattisgarh, Gujarat and Tamil Nadu) are implementing the scheme through mixed models involving both state agencies and insurance companies. Two districts were selected in each state (three districts in Bihar and Uttar Pradesh) to account for diverse performance levels within states.

Reviewer 3 Report

The planning of the implementation process is very useful and disputable. However, there several inter-dependencies in the practical applications that make the whole attempt uncertain. My recommendation is to try to model the dependence consequences.

Also, this paper does not consider the quantitative aspects of the insurance business. As result the data are not leading to some calculations that can suggest the level of the premiums and reserves that can make the whole enterprise sustainable. Further more the factors that are considered are not independent to make the probabilistic model easy. Therefore the risk of failure is rather high.

Author Response

Response to Reviewer 3 Comments

The planning of the implementation process is very useful and disputable. However, there several inter-dependencies in the practical applications that make the whole attempt uncertain. My recommendation is to try to model the dependence consequences.

Response 1: We do not understand the scope of this question and the relationship to the work we present in our manuscript. Our team is not responsible for the implementation process. We are exclusively responsible for the evaluation component and will do our best, given the constraints of the data available, to account for all possible sources of bias at the analytical level.

Also, this paper does not consider the quantitative aspects of the insurance business. As result the data are not leading to some calculations that can suggest the level of the premiums and reserves that can make the whole enterprise sustainable. Further more the factors that are considered are not independent to make the probabilistic model easy. Therefore the risk of failure is rather high.

Response 2: We wish to point out to the reviewer that the purpose of our analysis is not to assess the viability of the premium and financial sustainability of the scheme. Our focus is exclusively on exploring experiences of care, both from the provider and from the beneficiary point of view, under the implementation of the new insurance scheme in India, PM-JAY. Hence, our work does not attempt or even claims to attempt to investigate actuarial aspects of the scheme.

Round 2

Reviewer 3 Report

The study has marginal contribution in actuarial science. However, it can help the practitioners in some complex sistuations.